# Quality of the Diagnostic Process, Treatment Decision, and Predictors for Antibiotic Use in General Practice for Nursing Home Residents with Suspected Urinary Tract Infection

**DOI:** 10.3390/antibiotics10030316

**Published:** 2021-03-18

**Authors:** Stine Dyhl Sommer-Larsen, Sif Helene Arnold, Anne Holm, Julie Aamand Olesen, Gloria Cordoba

**Affiliations:** Research Unit for General Practice and Section of General Practice, Department of Public Health, 1014 Copenhagen, Denmark; stinedyhlk@gmail.com (S.D.S.-L.); siar@sund.ku.dk (S.H.A.); anneholm@sund.ku.dk (A.H.); aamand.julie@gmail.com (J.A.O.)

**Keywords:** urinary tract infection, elderly, nursing home, antibiotics, guidelines

## Abstract

Urinary tract infections (UTIs) are common in nursing home (NH) residents and Denmark is one of the countries with the highest antibiotic use in NHs. The aim of this study was to assess the quality of the diagnostic process and treatment decision on the day of the first contact from NHs to general practice and assess predictors for prescription of antibiotics in NH residents without an indwelling urinary catheter. The study was a prospective observational study in general practice in the Capital Region of Denmark; 490 patients were included; 158 out of 394 (40.1%, 95% CI 35; 45) patients with suspected UTI had urinary tract symptoms; 270 out of 296 (91.2%, 95% CI 87; 94) patients without urinary tract symptoms had a urine culture performed. Performing urine culture in the general practice was inversely associated to prescription of antibiotics on day one (OR 0.27, 95% CI 0.13; 0.56). It is imperative to support the implementation of interventions aimed at improving the quality of the diagnostic process on day one, as less than half of the patients given the diagnosis “suspected UTI” had urinary tract symptoms, and most patients without urinary tract symptoms had a urine culture performed.

## 1. Introduction

Antibiotic resistance is a growing problem and increases substantially due to the use, misuse, and overuse of antibiotics [1]. The Healthcare-Associated infections in Long-Term care facilities (HALT) project showed that, among the participating countries, Denmark had the highest antibiotic use for elderly residing in nursing homes (NHs) [2]. In Europe, urinary tract infection (UTI) is the most common reason for antibiotic prescribing in NH residents [2,3,4]. 

Estimates show that 24–80% of antibiotic use in NH residents is inappropriate [5,6,7,8]. High prevalence of asymptomatic bacteriuria and lack of an accepted clinical or laboratory gold standard to initiate treatment for UTI are among the most important reasons for the high prevalence of inappropriate use of antibiotics [9]. Consequently, consensus on the best diagnostic process to secure appropriate use of antibiotics in NH residents with suspected UTI does not exist. For example, there is wide variation in the combination of type and burden of symptoms that should be present to initiate empirical therapy. Furthermore, the use of urine dipstick is debated. Nonetheless, to avoid treatment of asymptomatic bacteriuria, most guidelines only recommended the performance of urine culture and antibiotic treatment in patients with symptoms related to the urinary tract [10,11,12,13].

The inclusion of urine culture in the diagnostic process of a suspected UTI results in a two-phase decision-making process [14]. The first phase encompasses the diagnostic process and treatment decision taken the day the patient contact general practice (henceforth, day one). On day one, the information collected throughout the diagnostic process includes: demographic information, signs and symptoms, and use of diagnostic tests such as dipstick and/or microscopy. The general practitioner (GP) makes a treatment decision based on the clinical history and available point-of-care test results. In Denmark, it is uncommon that the GP visits the NH for a suspected UTI and the majority of consultations are concluded over the phone or by e-mail. Typically, the GP will decide to treat empirically or delay the treatment decision until after day one. Following a delay, the GP can revise the diagnostic decision according to the development in symptoms and the result of the urine culture. If the GP decides to order a urine culture, the urine sample is taken at the NH and sent either to the practice or to the closest microbiology department. Due to differences in the diagnostic information and treatment decision on days one and two, this article focuses on day one. Hence, the aims of this study were (1) to assess the quality of the diagnostic process and treatment decision for NH residents with suspected UTI but without indwelling urinary catheter, (2) to assess predictors for prescription of antibiotics.

## 2. Results

### 2.1. Baseline Characteristics, Diagnostics, and Treatment Information

In total, 47 practices included 490 patients with suspected UTI without indwelling urinary catheter; 407 (83%) were women and the mean age was 84.6 (SD 8.7) years; 244 (53%) had dementia, and 110 (24%) had recurrent UTI (24%)—Table 1. The most common type of initial consultation between NH and general practice was telephone calls (260 calls (53%)) followed by face-to-face contacts (116 contacts (24%)). Dipstick testing was performed on 435 NH residents (89%) and urine culture in 406 NH residents (83%). In 394 of the patients (82%), UTI was suspected to be the cause of the symptoms. In 143 (29%) patients, antibiotics were prescribed on day one, and three patients (1%) were hospitalized. See Table 1.

### 2.2. The Quality of the Diagnostic Process and the Treatment Decision

Table 2 presents the quality indicators of the diagnostic process and treatment decision. The first three indicators evaluate the diagnostic process. The first shows the proportion of patients who had urinary tract symptoms of those diagnosed with UTI (40.1%, 95% confidence interval (CI) 35; 45). The second shows the proportion of patients who had a urine culture performed of those without urinary tract symptoms (91.2%, 95% CI 87; 94). The third shows the proportion of patients who had a urine culture performed of those diagnosed with UTI (92.1%, 95% CI 88; 94).

The last three indicators evaluate the treatment decision. The first shows the proportion of patients prescribed antibiotics or hospitalized of those with pyelonephritis symptoms (36.3%, 95% CI: 19–57). The second shows the proportion of patients prescribed antibiotics on day one of those with exclusively lower urinary tract symptoms (33%, 95% CI; 26–40). Finally, the third shows the proportion of patients prescribed antibiotics on day one of those without urinary tract symptoms (26.3%, 95% CI: 21–31).

### 2.3. Predictors of Antibiotic Prescriptions

The second aim investigated predictors for antibiotic prescribing. Demographic characteristics, type of first contact communication, and clinical information related to specific signs and symptoms were not associated with prescription of antibiotics. After adjusting for variables that showed a significant association in the univariate analysis, performing urine culture in the general practice was inversely associated to prescription of antibiotics on day one (OR 0.27, 95% CI 0.13; 0.56)—Table 3.

## 3. Discussion

### 3.1. Summary of Main Findings

In this group of NH residents seeking care in general practice due to suspected UTI, 82% were diagnosed with “suspected UTI”, and 29% were prescribed antibiotics on day one. Less than half of the patients diagnosed with “suspected UTI” had urinary tract symptoms, and the majority of patients without urinary tract symptoms had a urine culture performed. Furthermore, performance of urine culture in practice on day one was inversely related to prescription of antibiotics on day one.

### 3.2. Strength and Limitations

On one hand, the most important strength of the study is the representativeness of the elderly population as the data collection process was simple, hence securing the consecutive inclusion of patients. On the other hand, lack of representativeness of the practices recruiting the patients might be one of the main limitations. GPs who accepted to participate in the study might already have been interested in rational diagnosis and treatment of UTI. If so, the results may reflect a higher quality of the diagnostic process and treatment decision. Hence, widespread mislabeling and misdiagnosis of suspected UTI in elderly population could be even more prevalent than the results suggest. 

Another limitation is the lack of face and content validity of the quality indicators used to assess the quality of the diagnostic and treatment process. To our knowledge, there are not international or national validated quality indicators to assess the quality of the diagnostic process and treatment decision in the elderly population seeking care in general practice due to suspected UTI [15,16]. Therefore, based on evidence-based guidelines [10,11,12,13], we proposed and used a set of quality indicators that we plan to use as the base for the validation and consensus process in Denmark. 

Finally, we were only able to investigate the quality of the diagnostic process and treatment decision on day one, hence we could not assess whether the results of the urine culture led to appropriate or inappropriate prescription of antibiotics on day two. A recent study including 591 NHs [17] found that performance of urine culture was associated with inappropriate prescription of antibiotics on day two (i.e., over prescription), and higher risk for colonization of *C. difficile*. Our results showed that the performance of urine culture on day one was a protective factor for prescription of antibiotics; however, our results also showed that most patients without urinary tract symptoms had a urine culture performed. Thus, from our results, it is possible that significant bacterial growth in the urine culture might have been misinterpreted as UTI leading to over-prescription of antibiotics in asymptomatic patients when the result of the culture was available.

### 3.3. Comparison with Previous Literature and the Institute for Rational Pharmacotherapy (IRF) Guideline

On one hand, the IRF guideline by the Danish National Board of Health [12] recommends that in patients presenting exclusively with lower urinary tract symptoms, antibiotic treatment should be started only after receiving a positive urine culture. In this study, 67% of the patients presenting exclusively with lower urinary tract symptoms did not get a prescription on day one. It suggests that GPs generally postponed antibiotic prescription as recommended in the IRF guideline.

On the other hand, only about one third of patients with symptoms indicative of pyelonephritis received antibiotics on day one. The IRF guideline does not clearly recommend to start antibiotics in this type of patients, which should explain the low percentage given antibiotics on day one. Another explanation for this interesting finding is the debate about the definition of complicated UTI (e.g., pyelonephritis). Probably, for many GPs, the presence of urinary tract symptoms together with shaking chills or fever (our definition of pyelonephritis) did not necessarily mean that the patient had a complicated UTI. As debated in Johansen et al. [18], the definition of complicated UTI is very heterogeneous and not always clear. It is due to a large complex list of signs and symptoms as well as risk factors that leads to a heterogeneous assessment of the severity, consequently affecting the decision about prescribing antibiotics. 

Furthermore, our results are in line with previous studies reporting mislabeling of suspected UTI in patients without urinary tract symptoms, leading to inappropriate use of antibiotics. We showed that 60% of the NH residents who received the diagnosis UTI did not have any urinary tract symptoms. Previous studies have also shown a high percentage (41-70%) of antibiotic treatment for asymptomatic bacteriuria [5,19,20,21]. It can be explained by the fact that unspecific changes in the general condition of the elderly are interpreted by clinicians and other health care workers as potential indicators for suspected UTI. However, there is no robust evidence [6] showing a positive association between these unspecific findings and UTI.

Finally, in this study, 91% of patients without urinary tract symptoms had a urine culture performed. Other studies have shown that the majority of elderly in NHs do not have urinary tract specific signs or symptoms on the day the urine culture was ordered [22]. At the same time, studies have shown that the use of urine culture is associated with antibiotic prescription in NH residents [17,22]. We did not see this urine culture driven antibiotic use, probably because we only focused on day one.

### 3.4. Relevance

This study confirms that in a high-antibiotic use setting as Danish NHs, some of the major quality problems are inadequate indication for performance of a urine culture and prescription of antibiotics in patients displaying no urinary tract symptoms. High-quality-antibiotic stewardship programs aiming at reducing these inappropriate clinical practices are warranted.

## 4. Materials and Methods

### 4.1. Study Design

This study is a prospective observational study based on registration charts completed by general practices in the Capital Region of Denmark. 

### 4.2. Data Collection

All general practices in the Capital Region of Denmark received an invitation by letter or were recruited by advertising in a newsletter between January and March 2018. The data collection lasted from 1st of April to 30th of September 2018. Each general practice could register as many patients as wanted. 

Any person employed in the practice could register the data. The registration chart was filled out on day one and included background information, diagnostic process and treatment/management decision. For each registered contact and completed registration chart, the GPs were compensated with an amount corresponding to 10 min of work. The registration chart is depicted in Appendix A.

### 4.3. Inclusion and Exclusion Criteria

The study had two levels of inclusion criteria: the first, at practice level, and the second, on an individual level. For the first one, each general practice had to have enough elderly patients residing in NHs that at least five registered contacts during the registration period were feasible. For the second one, we took into consideration that our aim was to assess the quality of the diagnostic process and treatment decision. It means we needed to record the decisions taken during normal practice. Hence, the only instructions for inclusion of patients given to the GPs were that patients should be ≥65 years, residing in a NH, and the NH staff, the general practice staff, or the general practitioner should suspect UTI. Patients using a urinary catheter were excluded. 

### 4.4. Analysis

We defined the quality indicators based on previous literature addressing the quality of the diagnostic process in the elderly population seeking care in general practice due to suspected UTI [10,11,13] and the IRF guideline for the management of UTI in the elderly in Denmark [12]. Consensus among the authors was achieved discussing the content and relevance of the indicators in two meetings with subsequent iterative modification via e-mails. Urinary tract symptoms were defined as at least one of the following symptoms: dysuria, urgency, frequency, incontinence, suprapubic pain, costovertebral angle pain, and fever/shaking chills. 

Patients were classified as having lower UTI, if presenting with any combination of the following lower urinary tract symptoms (dysuria, urgency, frequency, incontinence, suprapubic pain).

For our second aim, to assess predictors for prescription of antibiotics, the statistical analysis was performed in two steps. First, possible predictors for the prescription of antibiotics on day one were investigated in a univariate hierarchical logistic regression model. We tested the following predictor variables individually with general practices as a cluster variable: a) anamnestic information, and b) use of point-of-care tests. The variables that were significant at the α < 0.05 were included in the multivariate analysis. All data work was performed in SAS Software version 9.4 (SAS Institute Inc., Cary, NC, USA).

### 4.5. Ethics

The study did not collect identifiable data at patient level and did not interfere with the normal treatment given to patients. The study did not require ethical approval by The Committee on Health Research Ethics of the Capital Region of Denmark cf. the Danish Committee Act §1 cl. 4 (Protocol number: 17031846).

## 5. Conclusions

Despite a relatively low antibiotic prescription rate on the first day of contact to general practice, the quality of the diagnostic process could be improved, as less than half of the patients who received the diagnosis “suspected UTI” had urinary tract symptoms. The study found an inverse association between performing urine culture in practice and prescription of antibiotics on day one, but most patients without urinary tract symptoms had a urine culture performed. This could indicate that interventions aimed at improving the knowledge of criteria for performing urine cultures are warranted. 

## Figures and Tables

**Table 1 antibiotics-10-00316-t001:** Baseline characteristics of the patients, diagnostics, and treatment information; *n* = 490.

Baseline Characteristics	*n* (%)
Women (*n* = 490)	407 (83)
^¥^ Age (*n* = 490)	84.6 (8.7)
Dementia (*n* = 459)	244 (53)
Recurrent UTI (*n* = 459)	110 (24)
Antibiotic prophylaxis for UTI (*n* = 459)	21 (5)
Consultation	
Type of initial consultation (*n* = 489)	
Telephone	260 (53)
Face-to-face	116 (24)
Email consultation	87 (18)
Other	26 (5)
* Presence of urinary tract symptoms (*n* = 490)	194 (40)
Diagnostic tests (*n* = 490)	
^α^ Dipstick	435 (89)
Dipstick at NH	275
Dipstick at general practice	259
Microscopy at general practice	31 (6)
^β^ Urine culture	406 (83)
Urine culture at general practice	209
Urine culture at microbiology department	210
Diagnosed with “suspected UTI” (*n* = 483)	394 (82)
Prescribed antibiotics (*n* = 486)	143 (29)
Hospitalized (*n* = 486)	3 (1)

^¥^ Mean (SD) * Dysuria, urgency, frequency, incontinence, suprapubic pain, costovertebral angle pain, fever/shaking chills. ^α^ (*n* = 99) NH residents, who had dipsticks performed both at the NH and at the general practice. ^β^ (*n* = 13) NH residents, who had a urine culture performed at the general practice and at the department of microbiology.

**Table 2 antibiotics-10-00316-t002:** Quality indicators: Appropriate diagnosis and treatment on day one.

	Nominator: Denominator	% (95% CI)
**Diagnostic Process**		
1: Number of patients with urinary tract symptoms */Number of patients diagnosed with UTI	158:394	40.1(35;45)
2: Number of patients who had a urine culture performed/Number of patients without urinary tract symptoms	270:296	91.2(87;94)
3: Number of patients who had a urine culture performed/Number of patients diagnosed with UTI (Excluded: hospitalized patients)	360:391	92.1(88;94)
**Treatment Decision**		
4: Number of patients prescribed antibiotics on day one OR hospitalized/Number of patients with symptoms of pyelonephritis ^¥^	8:22	36.3(19;57)
5: Number of patients prescribed antibiotics on day one/Number of patients with exclusively lower urinary tract symptoms ^∞^	57:172	33(26;40)
6: Number of patients without urinary tract symptoms prescribed antibiotics on day one /Number of patients without urinary tract symptoms (Excluded: hospitalized patients)	78:296	26.3(21;31)

* Urinary tract symptoms: dysuria, urgency, frequency, incontinence, suprapubic pain, costovertebral angle pain, fever/shaking chills. ^¥^ Symptoms of pyelonephritis: any combination of lower tract symptoms WITH costovertebral angle pain, AND/OR + fever/shaking chills ^∞^ Lower UTI symptoms: any combination of urinary tract symptoms WITHOUT costovertebral angle pain, AND/OR + fever/shaking chills.

**Table 3 antibiotics-10-00316-t003:** Predictors for prescription of antibiotics on day one.

Variable	Univariate Analysis		Multivariate Analysis	
	OR	95% CI	OR	95% CI
Sex (reference: men)	1.31	0.74; 2.31	N/A	N/A
Age	0.99	0.97; 1.02	N/A	N/A
Dementia (reference: no)	1.02	0.66; 1.56	N/A	N/A
Recurrent UTI (reference: no)	0.82	0.49; 1.36	N/A	N/A
Antibiotic prophylaxis for UTI (reference: no)	2.84	1.15; 6.99	3.58	1.08; 11.8
**Type of first contact**				
Written	Reference	Reference	Reference	Reference
Telephone	2.29	1.28; 4.11	1.7	0.82; 3.51
Face-to-face	1.38	0.66; 2.88	1.01	0.42; 2.42
**Signs & Symptoms**				
Dysuria (reference: no)	1.37	0.87; 2.16	N/A	N/A
Urgency (reference: no)	1.06	0.58; 1.9	N/A	N/A
Frequency (reference: no)	1.22	0.74; 2	N/A	N/A
New onset incontinence (reference: no)	0.72	0.14; 3.6	N/A	N/A
Suprapubic pain (reference: no)	2.28	0.94; 5.5	N/A	N/A
Macrohematuria (reference: no)	1.92	0.7; 5.26	N/A	N/A
No specific changes (reference: no)	1.2	0.79; 1.83	N/A	N/A
Fever/Shaking chills (reference: no)	1.2	0.59; 2.45	N/A	N/A
Flank pain (reference: no)	2.25	0.74; 6.84	N/A	N/A
**Diagnostic tools**				
Dipstick at NH	2.14	1.37; 3.35	1.26	0.72; 2.21
Dipstick at general practice	0.33	0.14; 0.56	0.5	0.24; 1
Microscopy at general practice	0.45	0.15; 1.32	N/A	N/A
Urine culture at general practice	***0.25***	***0.14; 0.45***	***0.27***	***0.13; 0.56***
Urine culture at microbiology department	2.45	1.53; 3.93	1.69	0.92; 3

N/A: not included in the multivariable hierarchical model.

## Data Availability

Data available under request.

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
