# Peer review of "Quality of the Diagnostic Process, Treatment Decision, and Predictors for Antibiotic Use in General Practice for Nursing Home Residents with Suspected Urinary Tract Infection"

_antibiotics, 2021, doi:10.3390/antibiotics10030316_

Round 1

Reviewer 1 Report

The Authors of this paper wanted to decribe th diagnostic process and quality and predictors of antibiotic prescription for UTI in resident in nursing home. Since the argument was interesting, it was not developed in a completely adequate manner. In particular, inclusion criteria were not completely clear. major concerns are:

  • inclusion criteria
  • definition of suspected UTI (only 194/490 patients had at least an UTI symptoms while for 435 patients a dipstick was performed)
  • quality indicators
  • evaluation at day 1 is a significant limitation of the quality of data for the purpose of the paper as specified also by the Authors

In addition to this, the discussion should be more extensive with more comments on the results, introduction should be shorten and tables restructured in a more readable way. Finally, appendix A could be removed because it did not represent significant additional information

Author Response

REVIEWER 1

Thank you for giving us the opportunity to submit a revised draft of the manuscript.  We would like to thank you as well for your relevant feedback. It has been very important to improve the content of the manuscript. In the following section you find a point-by-point response to your comments.

Reviewer 1

Comment

  1. Inclusion criteria
  2. Definition of suspected UTI (only 194/490 patients had at least an UTI symptoms while for 435 patients a dipstick was performed)
  3. Quality indicators
  4. The discussion should be more extensive with more comments on the results
  5. Introduction should be shorten and tables restructured in a more readable way. Finally, appendix A could be removed because it did not represent significant additional information

Answer

  1. Thank you for your comment. We have expanded the explanation of the inclusion criteria in the methods section to clarify the inclusion criteria at the individual level, which clarify as well your points about the definition of suspected UTI (next question).
  2. Thanks for pointing out that we need to clarify the definition of suspected UTI. Along the article we work with two definitions. The first one is the definition related to suspected UTI as an inclusion criteria. As the main objective of the study was to assess the quality of the diagnostic process and treatment decision during daily practice, the only instructions for inclusion of patients given to the GPs were that the patients should be ≥65 years, residing in a NH and the NH staff, the general practice staff, or the general practitioner should suspect UTI. (page 6 line 19). The second one is the definition to construct the quality indicators. As explained in the methods and discussion section, this definition was based on previous literature and guidelines addressing quality of the diagnostic process and management of suspected UTI in the elderly.
  3. Thank you for your comment on the quality indicators. As mentioned in the discussion section we have reflected about the lack of the validity of the quality indicators. We proposed the quality indicators based on evidence-based guidelines, which also forms the basis of the article, as there to our knowledge are no national or international validated quality indicators of the diagnostic process and treatment decision when suspecting UTI in the elderly.
  4. Thanks for you comment, we extended the discussion. For example we also discuss the low amount of patients with suspected pyelonephritis, who got antibiotics on day one. Besides the urine culture driven antibiotic prescription and the unspecific changes in the elderly as indicators of UTI.
  5. Thanks for your view on the setup. Despite the fact that the introduction seems long, we think that all information in the introduction is crucial to understand the reporting of the study. We now tried to make the tables more readable. As for the appendix we will leave it, as some readers would like to see the data collection instrument even though it does not represent new additional information.

Reviewer 2 Report

This is an observational study performed in Danish nursing home care setting. The aim of this study is to assess the quality of diagnosis and treatment and to identify factors associated with antimicrobial prescriptions for urinary tract infections in nursing home care residents. A total of 490 patients consulted at 47 GP practices between April and September 2018 were included in the study. Of 394 patients diagnosed with suspected UTI, 158 had urinary tract symptoms. Of the 296 patients without urinary tract symptoms, urine culture was obtained in 270 cases. Since only half of the patients diagnosed with suspected UTI were symptomatic and urine cultures were obtained in many of the patients without urinary tract symptoms, the authors concluded that improving the quality of diagnostic process and performing urine cultures for UTI is necessary.

Antibiotic is frequently overused in the nursing home. I think that this study clearly revealed the problem about the diagnostic and treatment practices for UTI for effective antibiotic stewardship program, but there are some points that need to be clarified, as follows:

Major comment:

  1. Although it is mentioned in the Introduction and Discussion section, I think it would be better to discuss more about the appropriateness of obtaining urine cultures for patients without urinary tract symptoms in the manuscript. Some clinicians may think that it is reasonable to obtain a urine culture even in the absence of urinary tract symptoms, because elderly patients with urinary tract infection sometimes present with atypical symptoms (PMID 24391677).

  2. It is surprising that the majority of patients included in this study did not have symptoms suggestive of a urinary tract infection, although they were suspected to have urinary tract infections by the NH staff, the general practice staff, or the general practitioner. Did the patients without symptoms suggestive of a urinary tract infection have other symptoms that led them to see their GP?

  3. In this study, more than half of the initial consultations between NH and general practice are done by telephone. Does this mean that there are cases where the indications for urine culture and antibiotic prescriptions are decided only by telephone? Please explain the practice system in Denmark more in detail. In addition, although the multivariate analysis did not show any significant difference, I feel it is desirable to mention in the discussion the possibility of difference in antibiotic prescription by practice style.

  4. It is surprising that more than half of the patients with symptoms of pyelonephritis did not receive antimicrobials on the first day. Please explain any possible reasons for the low prescription of antibiotics.

Minor comments:

  1. The number of practices listed in the abstract (47 practices) did not match the number listed in the result section (7 practices).

  2. The results of multivariate analysis about the factor associated with antibiotic prescription should be reported in the abstract since it is one of the aims in this study.
  3. There are two definitions of "suspected UTI" in the result section, which may be somewhat confusing.

  4. Please clarify how many GPs are there in Denmark to presume the selection bias in this study.

  5. Line 74 "The third shows the proportion of patients who had a urine culture performed of those without urinary tract symptoms" should be modified as follows: " The third shows the proportion of patients who had a urine culture performed of those diagnosed with UTI".

Author Response

Reviewer 2

Thank you for giving us the opportunity to submit a revised draft of the manuscript.  We would like to thank you as well for your relevant feedback. It has been very important to improve the content of the manuscript. In the following section you find a point-by-point response to your comments.

Comment

Major comments:

  1. Although it is mentioned in the Introduction and the Discussion section, I think it would be better to discuss more about the appropriateness of obtaining urine cultures for patients without urinary tract symptoms in the manuscript. Some clinicians may think that it is reasonable to obtain a urine culture even in the absence of urinary tract symptoms, because elderly patients with urinary tract infection sometimes present with atypical symptoms (PMID 24391677).
  1. It is surprising that the majority of patients included in this study did not have symptoms suggestive of a urinary tract infection, although they were suspected to have urinary tract infections by the NH staff, the general practice staff, or the general practitioner. Did the patients without symptoms suggestive of a urinary tract infection have other symptoms that led them to see their GP?
  2. In this study, more than half of the initial consultations between NH and general practice are done by telephone. Does this mean that there are cases where the indications for urine culture and antibiotic prescriptions are decided only by telephone? Please explain the practice system in Denmark more in detail. In addition, although the multivariate analysis did not show any significant difference, I feel it is desirable to mention in the discussion the possibility of difference in antibiotic prescription by practice style.
  1. It is surprising that more than half of the patients with symptoms of pyelonephritis did not receive antimicrobials on the first day. Please explain any possible reasons for the low prescription of antibiotics.

Minor comments:

  1. The number of practices listed in the abstract (47 practices) did not match the number listed in the result section (7 practices).
  1. The results of multivariate analysis about the factor associated with antibiotic prescription should be reported in the abstract since it is one of the aims in this study.
  2. There are two definitions of "suspected UTI" in the result section, which may be somewhat confusing.
  3. Please clarify how many GPs are there in Denmark to presume the selection bias in this study.
  4. Line 74 "The third shows the proportion of patients who had a urine culture performed of those without urinary tract symptoms" should be modified as follows: " The third shows the proportion of patients who had a urine culture performed of those diagnosed with UTI".

Answer

Major comments:

  1. We agree that this point is often misinterpreted in the clinical settings and it should be emphasized. We argue in the introduction that no gold standard for the diagnosis exists but most clinical guidelines agree that urine culture should not be obtained and treatment should not be issued unless the patient exhibits symptoms from the urinary tract. We have elaborated on this point in both the introduction and the discussion sections.
  2. We would like to highlight that one of the aims of the study was to assess the quality of the diagnostic process. Hence, we aimed at recording the diagnostic process and decision under real life conditions. It means the only instructions for inclusion of patients given to the GPs were that the patients should be ≥65 years, residing in a NH and the NH staff, the general practice staff, or the general practitioner should suspect UTI. We have added text in the discussion (page 5 lines 25-40) pointing out the challenges for defining UTI, the subsequent heterogeneity, and the lack of robust evidence of unspecified symptoms commonly associated with suspected UTI.
  3. Usually, the nursing home staff observes the patient and then contacts the general practice for further instructions. In Denmark, it is uncommon that the GP visits the NH for a suspected UTI and the majority of consultations are concluded over the phone or by e-mail. Typically, the GP will decide to either treat empirically or wait for the result of the urine culture. If the urine culture is positive the results are emailed to the NH with instructions for treatment. (Ref. Arnold et al 2020: Development of a Tailored, Complex Intervention for Clinical Reflection and Communication about Suspected Urinary Tract Infections in Nursing Home Residents). We have elaborated on this point in the introduction.
  4. It is indeed a surprising finding. We have added text in the discussion section to reflect on this finding (page 5, line 21).

Minor comments:

  1. Thank you, you are absolutely right. We have corrected it so that it appears with 47 general practices in both sections.
  2. Thanks for your comment. We have now mentioned it in the abstract.
  3. Thanks for pointing out that we need to clarify the definition of suspected UTI. Along the article we work with two definitions. The first one is the definition related to suspected UTI as an inclusion criteria. As the main objective of the study was to assess the quality of the diagnostic process and treatment decision during daily practice, the only instructions for inclusion of patients given to the GPs were that the patients should be ≥65 years, residing in a NH and the NH staff, the general practice staff, or the general practitioner should suspect UTI. (page 6 line 19). The second one is the definition to construct the quality indicators. As explained in the methods and discussion section, this definition was based on previous literature and guidelines addressing quality of the diagnostic process and management of suspected UTI in the elderly.
  4. Thanks for your comment. In 2020 there was 3326 general practitioners in Denmark. In this study only 47 of these general practices were included and as mentioned in the discussion we have reflected about the lack of representativeness of the GPs.
  5. Thank you very much, we have corrected it.

Reviewer 3 Report

The authors entitled their manuscript "Quality and predictors for antibiotic use in general practice for 2 nursing home residents with suspected urinary tract infection". However, although I found the manuscript interesting and adequately structured, it seems the main message stemming from the study is diagnostic and not therapeutic: many urine cultures were performed in patients without symptoms, possibly leading to inappropriate use of antibiotics after culture results. However, use of antibiotics after day 1 was not registered, which is major limitation of the manuscript (and the reason why I feel the current title may be inadequate).

Besides this, the manuscript is coherent, and the sample size is adequate allowing sufficiently narrow confidence intervals for generalization. I suggest to clearly identify primary and secondary objectives (and/or endpoints) in methods, and to shortening the manuscript (this is a descriptive study providing only descriptive findings that may be better suited for a brief report format).

Author Response

Reviewer 3

Thank you for giving us the opportunity to submit a revised draft of the manuscript.  We would like to thank you as well for your relevant feedback. It has been very important to improve the content of the manuscript. In the following section you find a point-by-point response to your comments.

Comment

  1. The authors entitled their manuscript "Quality and predictors for antibiotic use in general practice for 2 nursing home residents with suspected urinary tract infection". However, although I found the manuscript interesting and adequately structured, it seems the main message stemming from the study is diagnostic and not therapeutic: many urine cultures were performed in patients without symptoms, possibly leading to inappropriate use of antibiotics after culture results. However, use of antibiotics after day 1 was not registered, which is major limitation of the manuscript (and the reason why I feel the current title may be inadequate).
  2. Besides this, the manuscript is coherent, and the sample size is adequate allowing sufficiently narrow confidence intervals for generalization. I suggest to clearly identify primary and secondary objectives (and/or endpoints) in methods, and to shortening the manuscript (this is a descriptive study providing only descriptive findings that may be better suited for a brief report format).

Answer

  1. Thanks for your reflection on the title. We agree and have changed the title to: Quality of the diagnostic process, treatment decision and predictors for antibiotic use in general practice for nursing home residents with suspected urinary tract infection.
  2. Thank you for your comment. The two other reviewers ask for a longer discussion and shortening down the article in total is therefore not possible.

Round 2

Reviewer 1 Report

The Authors described the diagnostic process of UTI in NH residents and evaluated possible predictors of antibiotic prescription. The manuscript was improved after the previous revision although any comments were not completely answered (length of the introduction, improvement of tables) and the discussion section could be clearer.

In conclusion, I suggest to better discuss the result especially the inconsistent match between UTI symptoms and UTI diagnosis/suspected diagnosis. 
